# GelMA Hydrogel as a Promising Delivery System for Osthole in the Treatment of Rheumatoid Arthritis: Targeting the miR-1224-3p/AGO1 Axis

**DOI:** 10.3390/ijms241713210

**Published:** 2023-08-25

**Authors:** Weilin Zhang, Zhencong Li, Zhiwen Dai, Siyuan Chen, Weixiong Guo, Zhongwei Wang, Jinsong Wei

**Affiliations:** Department of Spinal Degeneration and Deformity Surgery, Affiliated Hospital of Guangdong Medical University, Zhanjiang 524001, China; weilin.zhang@gdmu.edu.cn (W.Z.); chancol@gdmu.edu.cn (Z.L.); zhiwen.dai@gdmu.edu.cn (Z.D.); siyuan.chen@gdmu.edu.cn (S.C.); weixiong.guo@gdmu.edu.cn (W.G.); zhongwei.wang@gdmu.edu.cn (Z.W.)

**Keywords:** rheumatoid arthritis, osthole, AGO1, GelMA

## Abstract

Rheumatoid arthritis (RA) is a multifaceted, chronic, progressive autoimmune disease. This study aims to explore the potential benefits of an enhanced drug delivery system utilizing optimized Gelatin Methacryloyl (GelMA) vectors in RA management. We evaluated the levels of miR-1124-3p and AGO1 in RA tissues and cell lines using qPCR, WB, and immunofluorescence. The effects of osthole on inflammatory response and joint morphology were determined by qPCR, H&E staining, and micro-CT. The data showed that miR-1224-3p was downregulated in RA tissues and HUM-iCell-s010RA cells, while the overexpression of miR-1224-3p in HUM-iCell-s010RA cells reduced the expression of IL-6 and IL-1β. Luciferase assay demonstrated that AGO1 was a direct target gene of miR-1224-3p. Additionally, osthole treatment increased miR-1224-3p levels and decreased AGO1 expression. The release data showed that osthole loaded on GelMA was released at a slower rate than free osthole. Further studies in a mouse model of CIA confirmed that osthole-loaded GelMA was more effective in attenuating osteopenia in RA as well as alleviating autoimmune arthritis. These findings suggest that osthole can regulate the miR-1224-3p/AGO1 axis in RASFs cells and has the potential to be developed as a clinical anti-RA drug. GelMA could provide a new approach to long-term RA treatment.

## 1. Introduction

Rheumatoid arthritis (RA) is a chronic autoimmune disease that primarily affects the joints, causing inflammation and joint damage [1,2,3]. Over the years, significant progress has been made in understanding the pathogenesis of RA and developing new therapies to treat it. One area of research that has gained attention in recent years is the relationship between RA and bone metabolism. Bone is a dynamic tissue that is continuously remodeled, with a balance between bone formation and bone resorption. In RA, the balance between these processes is disrupted, leading to bone destruction and the development of erosions [4]. This bone destruction can occur early in the disease process and is often irreversible, leading to significant disability and reduced quality of life [5].

Several factors contribute to the disruption of bone metabolism in RA. One of the key players is the cytokine receptor activator of nuclear factor kappa B ligand (RANKL) [6]. RANKL is a molecule that plays a critical role in the formation and activation of osteoclasts, the cells responsible for bone resorption [7,8]. In RA, RANKL is overexpressed, leading to increased osteoclast activity and bone destruction [9]. Targeting RANKL with biological agents has been shown to reduce bone erosion in RA patients.

MicroRNA (miRNA) is a small non-coding RNA (about 20 nt in length) that is able to regulate almost all known biological activities in organisms [10,11]. Several studies have identified dysregulated miRNAs in RA patients, with potential roles in regulating bone metabolism. For example, miR-146a, which is downregulated in RA, has been shown to inhibit RANKL-induced osteoclastogenesis [12,13]. miRNAs also can act as oncogenes or tumor suppressors [14,15]. Dysregulated miRNAs were related to hallmark regulation in tumors, including the maintenance of proliferative signaling, evasion of growth suppressors, resistance to cell death, activation of invasion and metastasis, and induction of angiogenesis [16]. Previous studies have indicated that miRNAs could be used as potential indicators for human cancer diagnosis, prognosis, and therapy [17,18]. miR-1224-3p plays an essential role in the development of tumors. miR-1224 regulates tumor proliferation, metastasis, invasion, angiogenesis, and drug resistance. In comparison, there is limited research specifically investigating the function of miR-1224-3p in RA.

In addition to the dysregulation of miRNAs, other mechanisms may contribute to the disruption of bone metabolism in RA. For example, inflammation in RA can lead to the production of pro-inflammatory cytokines, such as interleukin-6 (IL-6), which can stimulate osteoclastogenesis and inhibit osteoblast differentiation [19]. Osthole, also known as osthol, a natural coumarin extracted from the Cnidium monnieri and Angelica pubescens, has been shown to have anti-inflammatory, antioxidant, and anticancer activities [20]. Regarding anti-tumor, osthole has inhibitory effects on breast cancer [21], ovarian cancer [22], and liver cancer [23] by inhibiting cancer cell proliferation, invasion, migration, inhibiting cancer angiogenesis, and inducing cancer cell apoptosis [24]. At the same time, osthole showed antioxidant and moist effects by intervening NF-kB and p38 MAPK pathways [25]. But whether osthole has the effect of inhibiting RA progression needs further study.

At present, the targeting, safety, and efficacy of rheumatoid arthritis drugs cannot meet the needs of clinical treatment. Therefore, the application of new biomaterials has become the key to filling the existing shortcomings. GelMA is a covalently cross-linked hydrogel and a hydrolysate of collagen. GelMA has better solubility and lower antigenicity than collagen [26,27] and has been used in orthopedics [28] and other fields [29] for many years. GelMA exhibits better biocompatibility, good cell viability, and response to cells and has been widely used in various biological applications [30]. Its outstanding drug-release kinetic properties and bioelasticity make it an ideal material for many medical applications [30], including cell culture scaffolds, tissue engineering platforms, and drug, gene, or growth factor delivery carriers [29,31,32].

RA is an autoimmune disease characterized by chronic and progressive symptoms. Managing the inflammatory response of synovial fibroblasts is a crucial part of treating RA. In this study, we aim to explore the potential impact of an improved drug delivery system using optimized GelMA vectors in the treatment of RA.

## 2. Results

### 2.1. miR-1224-3P Was Downregulated in Rheumatoid Arthritis Synovial Tissues 

Synovial tissue samples were collected from 52 patients undergoing joint replacement surgery at the Affiliated Hospital of Guangdong Medical University between March 2021 and January 2022. Of these patients, 37 had primary knee osteoarthritis (OA), and 15 had rheumatoid arthritis (RA). The synovial tissue was extracted from the knee joint and stored in liquid nitrogen. All patients provided informed consent and were treated in accordance with the ethical regulations of the Affiliated Hospital of Guangdong Medical University. Healthy synovial tissue cannot be obtained through surgery, and the 37 patients with knee osteoarthritis comprised the control group. Firstly, we conducted a qPCR analysis of synovial tissue, revealing that the IL-6 level was significantly elevated in the RA group compared to the OA group (Figure 1A). Furthermore, we assessed the levels of the target miR-1224-3p, observing a significant reduction in its expression within the RA group compared to the OA group (Figure 1B).

### 2.2. AGO1 Negatively Correlated with miR-1224-3p

Bioinformatics analysis (http://mirdb.org/mirdb/index.html, accessed on 19 May 2022) indicated that miR-1224-3p may target AGO1 (Figure 2A). Subsequently, qPCR was performed to measure the AGO1 mRNA levels, which were found to be significantly elevated in the RA group (Figure 2B). These results suggest that miR-1224-3p and AGO1 may play a role in rheumatoid arthritis. To further verify this, we conducted experiments on Human Primary Rheumatoid Arthritis Synovial Fibroblasts Cells (HUM-iCell-s010RA) cells. Briefly, wild-type (WT) and mutated (Mut) AGO1 3′UTRs were co-transfected with miR-1224-3p mimic. The results demonstrated that miR-1224-3p significantly decreased the luciferase activity of WT AGO1 in HUM-iCell-s010RA cells (Figure 2C), and our data indicated that AGO1 was a direct target gene of miR-1224-3p. Moreover, we conducted experiments on Human Primary Synovial Fibroblasts Cells (HUM-iCell-s010) and HUM-iCell-s010RA cells to analyze the level of miR-1224-3p and AGO1. As shown in Figure 2D and E, miR-1224-3p was significantly decreased in HUM-iCell-s010RA cells, while the mRNA levels of AGO1 were significantly increased. Additionally, we also measured the levels of IL-6 (Figure 2F) and IL-1β (Figure 2G), and our data showed that both IL-6 and IL-1β were significantly higher in HUM-iCell-s010RA cells compared to HUM-iCell-s010 cells (*p* < 0.01). Furthermore, transfection of miR-1224-3p mimic in HUM-iCell-s010RA cells resulted in significant inhibition of AGO1 expression (Figure 2H,I). Our results reveal that AGO1 levels are significantly increased in RA and are negatively correlated with miR-1224-3p.

### 2.3. Osthole Treatment in HUM-iCell-s010 Rheumatoid Arthritis Cells

Osthole is a major bioactive compound extracted from Cnidium monnieri. Extensive studies have recognized its multiple activities [20], with anti-inflammatory, antioxidant, and anticancer activities. In our preliminary examination, HUM-iCell-s010 RA cells were treated with 50 μM of osthole. We found a significant decrease in the levels of IL-6 (Figure 3A) and IL-1β (Figure 3B) following osthole treatment for 24 h. Furthermore, we evaluated the changes in miR-1224-3p and AGO1 levels. Our results demonstrated that osthole treatment elevated the expression of miR-1224-3p (Figure 3C) and inhibited the expression of AGO1 in both mRNA (Figure 3D) and protein levels (Figure 3E). To investigate the effect of osthole on cell proliferation, we treated HUM-iCell-s010 RA cells with 50 μM of osthole and conducted an MTT assay to analyze cell proliferation. The results showed that osthole treatment significantly reduced proliferation (Figure 3F).

### 2.4. Immunofluorescence for Assessment of Osthole and Osthole-Loaded GelMA on HUM-iCell-s010 Rheumatoid Arthritis Cells

In our treatment of HUM-iCell-s010 RA cells using 50 μM of osthole and osthole-loaded GelMA, we assessed changes in AGO1 protein levels using immunofluorescence (Figure 4B). Our results demonstrated that osthole decreased the expression level of AGO1 in RA cells, while osthole-loaded GelMA further down-regulated the expression level of AGO1 (Figure 4A).

### 2.5. Property Characterization and Release Experiments of Osthole, GelMA, Osthole-Loaded GelMA

RA is a chronic condition that necessitates extended therapy, with the expense of drug treatment being a major concern. Enhancing the effectiveness of drugs may alleviate this issue considerably. Several studies have indicated that GelMA may boost drug efficacy. To clarify the nature and characterization of GelMA, osthole-loaded GelMA, we observed that GelMA had a loose porous structure using scanning electron microscopy, while osthole could fill these pores (Figure 5A,B,E). The osthole-loaded GelMA has a higher absorption peak at different wavelengths (Figure 5D). We test the drug release rate of osthole when administered either freely or loaded onto GelMA. As shown in Figure 5C, our results showed that nearly 100% of free osthole was released within 40 h, while osthole-loaded onto GelMA exhibited a release rate of approximately 48.2%. These findings confirm the potential of GelMA in controlling drug release rates. Due to its chronic nature, RA requires prolonged treatment cycles. Therefore, GelMA may prove to be a more advanced option for RA patients, as it offers better control over slow drug release. These results suggest that the use of GelMA as a drug carrier may enhance the efficacy of RA treatment and potentially reduce the burden of extended drug therapy.

### 2.6. Osthole-Loaded GelMA Shows Better Therapeutic Potential in CIA Mice Model

Subsequently, we utilized GelMA to deliver osthole to CIA mice to gauge its effects. Osthole was administered intraperitoneally to mice from day 10 of the second CIA induction, with PBS serving as the negative control (Figure 6A). All mice were sacrificed on day 52, and the hind ankle joints were preserved after the removal of the soft tissue. Subsequently, the joints were fixed, decalcified, embedded in paraffin, and sectioned for histological examination using H&E staining. Our results indicate that joint inflammation was significantly reduced in the osthole-loaded GelMA groups compared to the free osthole group (Figure 6B, top row). Furthermore, the micro-CT analysis revealed that osthole-loaded GelMA had a positive impact on bone microstructure, with the greatest improvement observed in the osthole-loaded GelMA group (Figure 6B, bottom row). The transcription levels of IL-6 (Figure 6C) and IL-1β (Figure 6D) were downregulated in both the free osthole group and the osthole-loaded GelMA group. However, the decrease in IL-6 and IL-1β levels was significantly greater in the osthole-loaded GelMA group (*p* < 0.01). Furthermore, qPCR confirmed the upregulation of miR1224-3p mediated by osthole and the downregulation of AGO1 in all osthole treatment groups (Figure 6E,F). However, compared to osthole treatment, the groups treated with osthole-loaded GelMA demonstrated a significant increase in miR-1224-3p levels and a significant decrease in AGO1 mRNA levels. (*p* < 0.01). Correspondingly, osthole-loaded GelMA also exhibited the least inflammation response. These results demonstrate the greater efficacy of osthole-loaded GelMA in mitigating autoimmune arthritis in the CIA mouse model.

### 2.7. Effects of Osthole-Loaded GelMA in the Hip of CIA Mice Model

We still used the control group, CIA group, and CIA + osthole-loaded GelMA group of the mice described above. In all mice sacrificed at day 52, the upper femur, including the femoral head, was preserved after soft tissue removal. Subsequently, the joints were fixed, decalcified, and embedded in paraffin, and some femoral heads were stained with H&E for histological examination. The H&E-stained histology score used the Mankin score commonly used in previous studies [33,34,35] (Appendix A). Our results revealed that the damage to the articular end of the femoral head was evident in the CIA group (RA model), in contrast to which the joint damage was alleviated in the osthole-loaded GelMA group (Figure 7).

We finally performed a micro-CT examination using part of the upper femur, and scan assessment revealed that the number and density of bone trabecula in the femoral head of mice in the experimental group using intraperitoneal injection of osthole-loaded GelMA were higher than those of mice in the CIA group (RA) (Figure 8B). Similarly, the osthole-loaded GelMA group had better performance in bone mineral density (BMD), bone volume (BV/TV), bone thickness (Tb.th), and other values (Figure 8A).

## 3. Discussion

Rheumatoid arthritis (RA) is a chronic autoimmune disease that affects the joints and can lead to severe pain, stiffness, and functional disability. The current treatment for RA involves the use of disease-modifying antirheumatic drugs (DMARDs) and nonsteroidal anti-inflammatory drugs (NSAIDs). The dilemma in RA treatment lies in the fact that while these medications can help control symptoms and slow disease progression, they do not provide a cure. Additionally, they can have side effects, and not all patients respond equally to them. Furthermore, the treatment of RA typically requires long-term use of medication, which can lead to issues such as medication adherence and cost.

The pathogenesis of RA is complex and involves various molecular mechanisms, including the dysregulation of microRNAs (miRNAs), small non-coding RNAs that regulate gene expression by binding to target mRNAs. miRNAs play a critical role in the regulation of various biological processes, including inflammation, cell proliferation, and apoptosis, and their dysregulation has been implicated in the development and progression of RA. Our study found that miR-1224-3P was downregulated in RA synovial tissues compared to OA synovial tissues (Figure 1B). This finding suggests that miR-1224-3P may play a role in RA pathogenesis and could be a potential therapeutic target for the disease. Through bioinformatics analysis, we identified AGO1 (a member of the Argonaute family of proteins) as the potential target of miR-1224-3p. The study found that AGO1 was negatively correlated with miR-1224-3P (Figure 1 and Figure 2). Osthole is a major bioactive compound extracted from Cnidium monnieri, and it showed a strong anti-inflammatory property. Our study found that osthole could increase miR-1224-3P and decrease AGO1 expression in HUM-iCell-s010 RA cells, as well as reduce IL-6 and IL-1β levels in these cells (Figure 3 and Figure 4). This finding suggests that osthole may have therapeutic potential in RA by modulating miR-1224-3P and AGO1 expression, as well as reducing pro-inflammatory cytokine levels.

GelMA is widely recognized because of its biocompatibility and high solubility, making it a popular material in various industries. Its unique properties, including bioelastic and kinetic characteristics, make it an ideal material for many medical applications, such as drug delivery systems, cell culture scaffolds, etc. To enhance the utilization of osthole, we loaded it onto GelMA. Our assessment of the drug release of these delivery systems revealed that GelMA exhibited superior performance (Figure 5). Therefore, GelMA may be a more advanced option for RA patient administration. In the CIA mice model, we investigated the therapeutic potential of osthole-loaded onto GelMA and free osthole. The results showed that osthole-loaded GelMA showed better therapeutic potential in reducing joint inflammation and anti-osteopenia compared to free osthole (Figure 6, Figure 7 and Figure 8). Furthermore, the expression of the pro-inflammatory cytokines, IL-6 and IL-1β, was significantly reduced in the osthole-loaded GelMA group, indicating that GelMA could be a more effective drug delivery vehicle for osthole in the treatment of RA. In conclusion, Gelatin Methacryloyl (GelMA) is a versatile material that has promising applications in the medical field. Our study demonstrates that GelMA may be a superior option for drug delivery in RA treatment due to its high hydrophilicity and superior performance.

## 4. Materials and Methods

### 4.1. Clinical Specimens 

Synovial tissue samples were collected from 52 patients (37 with primary knee osteoarthritis and 15 with rheumatoid arthritis) undergoing joint replacement surgery at the Affiliated Hospital of Guangdong Medical University between March 2021 and January 2022. The synovial tissue was extracted from the knee joint and stored in liquid nitrogen. All patients had not received treatment such as radiation therapy or chemotherapy before the sample was collected. All patients signed the informed consent forms. This study was approved by the ethical committee of the Affiliated Hospital of Guangdong Medical University.

### 4.2. Cell Lines 

Human Primary Synovial Fibroblasts Cells (HUM-iCell-s010) and Human Primary Rheumatoid Arthritis Synovial Fibroblasts Cells (HUM-iCell-s010RA) were obtained from iCell Bioscience (Shanghai, China). The two cell lines were grown in Dulbecco′s Modified Eagle′s Medium (DMEM) (Thermo Fisher Scientific Inc., Waltham, MA, USA) supplemented with 20% fetal bovine serum (FBS) and 1% Penicillin-Streptomycin solution (Thermo) at 37 °C with 5% CO_2_. Cells were expanded up to 3 passages for further use.

### 4.3. Preparation and Characterization of Osthole-Loaded GelMA

#### 4.3.1. Preparation of GelMA

A 30 g amount of gelatin (Macklin, Shanghai, China) and 300 mL of PBS were thoroughly mixed and stirred for 2–4 h to completely dissolve the gelatin, and then fully stirred again in a water bath at 60 °C until it was completely dissolved and swelled. A microsyringe pump was used to slowly add 16 mL of methacrylic anhydride (MCE, Shanghai, China) at a rate of 0.25 mL/min, and it was reacted for 2 h. An 800 mL aliquot of PBS was added to stop the reaction. Then, the GelMA solution was transferred to a dialysis bag and dialyzed in deionized water at 38° for 3–4 days. Finally, the GelMA was frozen and dried using a freeze dryer [36].

#### 4.3.2. Prepare Osthole-Loaded GelMA Hydrogel

To fabricate the injectable osthole-loaded laponite-GelMA hydrogel, GelMA was first completely dissolved in DI water. Thereafter, osthole (50 mM) (MCE, Shanghai, China) was added to the GelMA solution. Irgacure 2959 (0.5% *w*/*v*) (Sigma, Shanghai, China) was added into the pre-polymer as a photoinitiator, which permitted the pre-polymer to cross-link under UV radiation (6.9 mW/cm^2^, 360–480 nm).

### 4.4. Osthole, GelMA Characterizations Assay

GelMA and osthole-loaded GelMA surface states were analyzed by SEM (Gemini 2, Zeiss, Oberkochen, Germany), absorption peaks by Fourier transform infrared spectroscopy (FTIR, INVENIO-R, Bruker, Ettlingen, Germany), porosity by SkyScan-1176 micro-CT (Bruker micro-CT, Bruker Belgium SA, Kontich, Belgium) system, CTAn software (version 1.17.7.2).

### 4.5. Transfection

HUM-iCell-s010RA cells were transfected with miR-1224-3p mimics (5′-CCCCACCUCCUCUCUCCUCAG-3′) or miR-NC (5′-CAGUACUUUUGUGUAGUACAA-3′) by using Lipofectamine 2000. Briefly, cells were cultured to 80% to 90% confluence after being seeded into 6-well plates. They were transfected with Lipofectamine 2000 according to the manufacturer′s instructions. miR-1224-3p mimics, or miR-NC (100 pmol in 10 μL) in 250 μL of Opti-MEMⅠ reduced serum medium were mixed with 500 μL of Lipofectamine 2000 transfection reagent dissolved in 250 μL of the same medium and were allowed to stand at room temperature for 20 min. The resultant 500 μL of transfection solutions were then added to each well. After 6 h, the cultures were replaced with 2 mL DMEM medium supplemented with 20% FBS and 1% Penicillin-Streptomycin solution

### 4.6. Quantitative Real-Time Polymerase Chain Reaction (RT-qPCR)

The cellular and tissue RNAs were extracted by using RNAiso Plus (TaKaRaBio Technology, Tokyo, Japan). The first-strand cDNA was synthesized by the Prime Script TM RT Master Mix (TaKaRaBio Technology, Tokyo, Japan) with random primers. Real-time PCR was carried out using the ABI 7900 Real-Time System (Applied Biosystems, Foster City, CA, USA) with TransStart^®^ Green qPCR SuperMix (TransGene, Beijing, China). The amplification program contained the initial denaturation at 95 °C for 30 s, followed by 45 cycles of denaturation at 95 °C for 5 s and extension at 60 °C for 30 s. The mRNA relative levels were measured by the 2^−ΔΔCt^ method with GAPDH and U6 as endogenous controls for AGO1 and miR-1224-3p, respectively. The sequences of the primers were as follows: hIL-1β-F: 5′-CATGGGATAACGAGGCTTATGT-3′; hIL-1β-R: 5′-CATATGGACCAGACATCACCAA-3′; hIL-6-F: 5′-CACTCACCTCTTCAGAACGAAT-3′; hIL-6-R: 5′-GCTGCTTTCACACATGTTACTC-3′; hAGO1-F: 5′-ACAGTGTCGAGAAGAGGTGCTC-3′; hAGO1-R: 5′-GAGTAGGTGTTCTTGAGATGCCG-3′; hGAPDH-F: 5′-GTCTCCTCTGACTTCAACAGCG-3′; hGAPDH-R: 5′-ACCACCCTGTTGCTGTAGCCAA-3′; hU6-F: 5′-CTCGCTTCGGCAGCACAT-3′; hU6-R: 5′-TTTGCGTGTCATCCTTGCG-3′; mIL-1β-F: 5′-TGGACCTTCCAGGATGAGGACA-3′; mIL-1β-R: 5′-GTTCATCTCGGAGCCTGTAGTG-3′; mIL-6-F: 5′-CTTCCATCCAGTTGCCTTCT-3′; mIL-6-FR: 5′-CTCCGACTTGTGAAGTGGTATAG-3′; mAGO1-F: 5′-CTGCCTTCTACAAAGCACAGCC-3′; mAGO1-R: 5′-TCTGTCCACAGTGGGTCACTTC-3′; mGAPDH-F: 5′-CATCACTGCCACCCAGAAGACTG-3′; mGAPDH-R: 5′-ATGCCAGTGAGCTTCCCGTTCAG-3′.

### 4.7. Dual-Luciferase Reporter Assay

The fragments of AGO1 3′UTR encompassing the miR-1224-3p-pairing region or miss-matched seed sequence were individually subcloned into the pMIR-REPORT vector downstream of the luciferase gene by Fubio (Fubio, Suzhou, China). HUM-iCell-s010RA cells (1 × 10^5^ cells/well) were co-transfected with 200 ng of the indicated reporter constructs, 50 ng of pRL-TK plasmid (Promega, Madison, WI, USA), and 20 nM of miRNA mimic. Luciferase activity was analyzed after 24 h using the Dual-Luciferase Reporter Assay System as recommended by the manufacturers (Promega).

### 4.8. Western Blot

The cellular proteins were extracted with cell lysis buffer (Invitrogen, Carlsbad, CA, USA). Protein samples were separated on 10% SDS-PAGE, followed by being transferred onto PVDF membranes (Millipore, Burlington, MA, USA). The membranes were blocked with 2% bovine serum albumin (BSA) in PBST for 1 h at room temperature; then, the membranes were incubated with the primary antibodies and secondary antibodies for 1h at 37 °C. Finally, the membrane was observed by sensitive enhanced chemiluminescence (ECL) detection kit (Beyotime, Shanghai, China), and bands were detected by ChemiDoc Imaging Systems (Bio-Rad, Hercules, CA, USA). Antibodies were used, including rabbit anti-AGO1 antibody (1:1000, Cell Signaling, Danvers, MA, USA) and mouse anti-β-actin antibody (1:1000, Cell Signaling). Secondary antibodies included HRP-conjugated goat anti-rabbit IgG antibody and anti-mouse IgG antibody (1:2000, Southern Biotech, Birmingham, AL, USA), HRP-conjugated donkey anti-rabbit IgG (H+L) antibody (1:2000, Santa Cruz, Dallas, TX, USA), and HRP-conjugated goat anti-rabbit IgG antibody (1:1000, Abbkine, Wuhan, China).

### 4.9. MTT Assay

Cells were seeded into cell wells and incubated for 48 h. Then, 50 μM of osthole or DMSO was used to incubate cells for another 24 h. To determine the cell viability, the culture media was changed, and then 50 µL of MTT dye was added. After 4 h incubation, the supernatant was discarded. Then, 200 µL of DMSO (dimethyl sulfoxide) solution was added to dissolve the formazan crystals. Finally, 30 µL of the solution from each well was transferred into 96 well plates. The absorbance was read at 570 nm by using Epoch 2 microplate reader.

### 4.10. Histological Evaluation

The samples were decalcified in a 10 wt% ethylenediaminetetraacetic acid (EDTA) solution, dehydrated through a series of graded ethanol, embedded in paraffin, and cut through a vertical section into 2 μm-thick sections. And then, the sections were stained with hematoxylin and eosin (H&E).

### 4.11. Immunofluorescence Staining

Synovial fibroblasts were fixed in 4% paraformaldehyde and embedded in paraffin. It was then heated in an oven at 60 °C for 1 h, dewaxed in dimethyl benzene, polarized with descending concentrations of alcohol (100, 95, and 75%), and washed with water. Endogenous peroxidase activity was prevented by incubation at room temperature with a 3% H_2_O_2_ solution. The antigenic epitope was blocked by incubation with 20% normal goat serum for 1 h, followed by incubation with primary antibodies at 4 °C for 12 h. After washing with water three times, the slides were incubated with an Alexa powder 488- or 594-conjugated secondary antibody (1:1000) for 1 h at room temperature. Then, the nuclei were dyed using a DAPI solution, and counterstaining was performed with hematoxylin.

### 4.12. Microstructure Analysis

Knee joints were fixed in 4% paraformaldehyde for 48 h and then washed with phosphate-buffered saline (PBS) and soaked in 75% ethanol. The SkyScan-1176 micro-CT (Bruker micro-CT, Bruker Belgium SA, Kontich, Belgium) system was used to scan the microstructural changes of knee joints. Micro-CT was performed on a 1 mm region of metaphyseal spongiosa in the distal femur, 0.5 mm above (femur) the growth plate, upper end of the femur, to evaluate the trabecular bone. The images were reconstructed and realigned in 3D using NR econ and CTAn software(version 1.17.7.2).

### 4.13. Induction of CIA and Treatment

Female DBA1/J mice were injected intradermally with 0.1 mL of the CII/CFA emulsion on day 0. Then, mice received intradermally with 0.1 mL of 2 mg/mL CII emulsified in incomplete Freund′s adjuvant (IFA; Chondrex, Woodinville, WA, USA) (2 mg/mL) on day 21. Osthole (MCE, Shanghai, China) 30 mg/kg [37], the same effect equivalent to osthole-loaded GelMA, was intraperitoneally injected on day 31, and different experimental groups were recorded. Synovial tissue and femurs were isolated from mice after sacrifice on day 52.

### 4.14. Statistical Analysis

All data were graphed, and statistical analyses were performed using Prism 9 software (version 9.0, GraphPad, La Jolla, CA, USA) and SkyScan-1176 micro-CT (Bruker micro-CT, Bruker Belgium SA, Kontich, Belgium) system. Comparisons between the two groups′ means were performed with a two-tailed Student′s *t*-test, whereas multiple comparisons were conducted by one-way analysis of variance (ANOVA). The results are presented as mean ± standard deviations (SD). The difference was considered to be statistically significant at *p* < 0.05.

## 5. Conclusions

In summary, our study highlights the therapeutic potential of miR-1224-3p, AGO1, and osthole in the treatment of RA. We also suggest that a GelMA drug delivery system could be developed to enhance the efficacy of osthole in RA treatment. However, further research is required to elucidate the underlying mechanisms of action. Our findings offer a promising new approach to long-term RA treatment, which could significantly improve the quality of life for millions of people worldwide. The development of GelMA-based drug delivery systems has the potential to revolutionize RA therapy, as it offers a sustainable, biodegradable, and biocompatible solution to drug administration. With further investigation, the GelMA delivery system could provide a more effective, safe, and convenient treatment option for RA patients, ultimately leading to better health outcomes and improved quality of life.

## Figures and Tables

**Figure 1 ijms-24-13210-f001:**
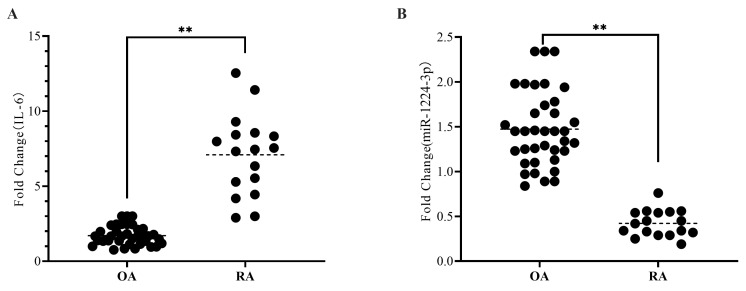
miR-1224-3P was downregulated in RA synovial tissues. The levels of IL-6 (**A**) and miR-1224-3p (**B**) in synovial tissue with osteoarthritis (OA) and rheumatoid arthritis (RA) patients were detected by RT-qPCR. ** *p* < 0.01.

**Figure 2 ijms-24-13210-f002:**
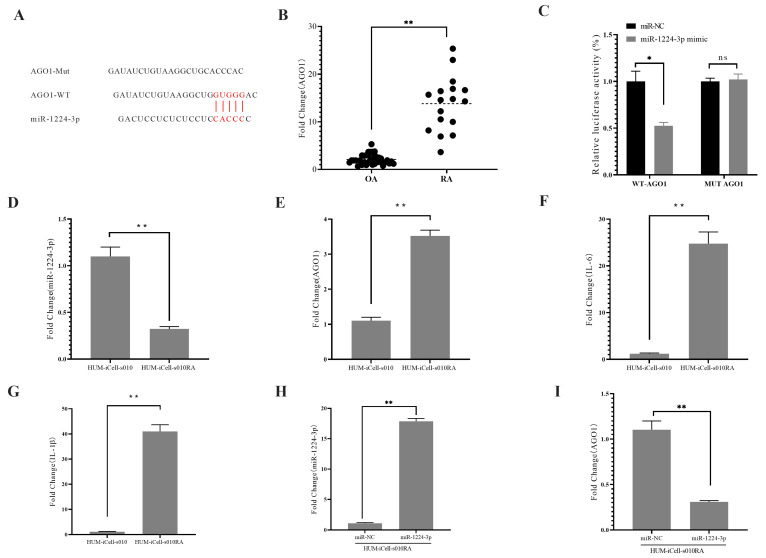
AGO1 negatively correlated with miR-1224-3p. (**A**) Schematic of the miR-1223-3p-pairing region and miss-matched target sequence within AGO1 3′UTR. Red font and lines indicate AGO1 has a specific binding region to miR-1223-3p. (**B**) Relative levels of AGO1 measured by qPCR in OA and RA synovial tissue. (**C**) Luciferase readout from HUM-iCell-s010RA cells co-transfected with WT-AGO1 3′UTR or MUT-AGO1 3′UTR and miR-1224-3p mimic or miR-NC mimic. Relative levels of miR-1224-3p (**D**), AGO1 (**E**), IL-6 (**F**), and IL-1β (**G**) were measured by qPCR in HUM-iCell-s010 and HUM-iCell-s010RA cells. The expression levels of miR-1224-3p (**H**) and AGO1 (**I**) were analyzed by qRT-PCR in transfected cells HUM-iCell-s010RA cells. All statistical data are represented as mean ± SD (*n* = 3, * *p* < 0.05, ** *p* < 0.01, ns: difference not significant).

**Figure 3 ijms-24-13210-f003:**
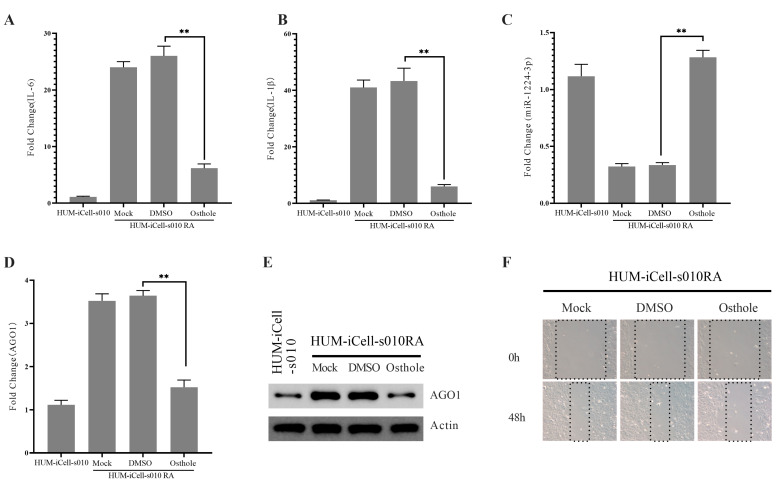
Osthole treatment in HUM-iCell-s010 RA cells. The relative levels of IL-6 (**A**), IL-1β (**B**), miR-1224-3p (**C**), and AGO1 (**D**) were quantified using qPCR in HUM-iCell-s010RA cells treated with osthole for 24h. The AGO1 protein levels in different treatments were evaluated through WB (**E**), while cell proliferation was analyzed using the MTT assay (**F**). All statistical data are represented as mean ± SD (*n* = 3, ** *p* < 0.01).

**Figure 4 ijms-24-13210-f004:**
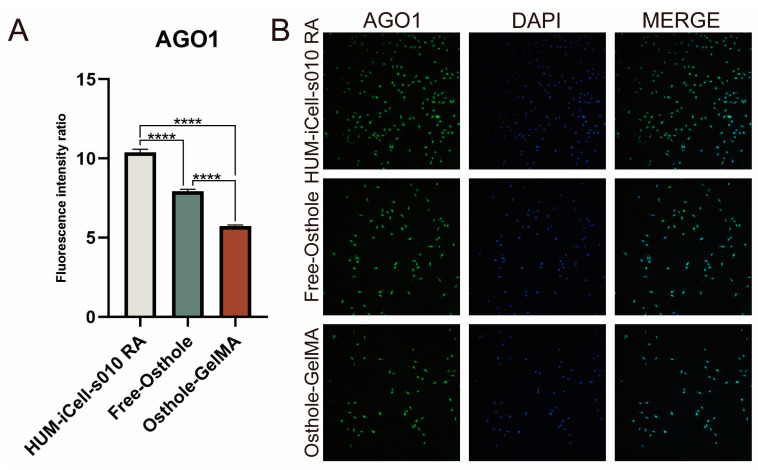
Immunofluorescence for assessment of osthole and osthole-loaded GelMA on HUM-iCell-s010 RA cells. (**A**) AGO1 protein fluorescence intensity differences between different experimental groups. (**B**) Immunofluorescence assay for AGO1 protein between different experimental groups (10×). All statistical data are represented as mean ± SD (*n* = 3, **** *p* < 0.001).

**Figure 5 ijms-24-13210-f005:**
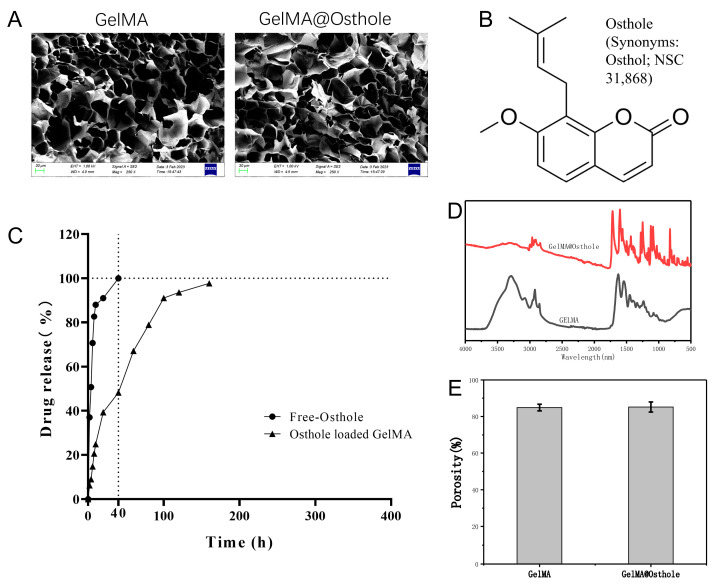
Characterization of osthole, GelMA, and osthole-loaded GelMA properties. (**A**) Morphology of GelMA and osthole-loaded GelMA under scanning electron microscope. (**B**) Molecular structure of osthole. (**C**) In vitro release profile of free osthole and osthole-loaded GelMA. (**D**) Fourier transforms infrared spectrometer scanning GelMA, osthole-loaded GelMA. (**E**) The porosity of GelMA and osthole-loaded GelMA was analyzed by CTAn. All statistical data are represented as mean ± SD (*n* = 3).

**Figure 6 ijms-24-13210-f006:**
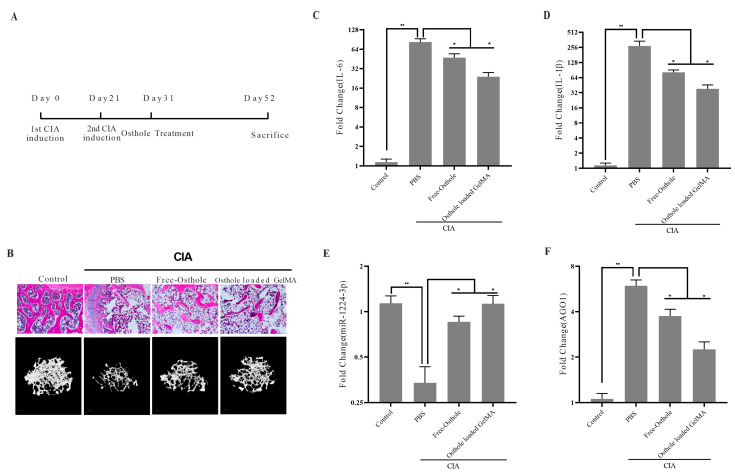
OstholeFig.6 Osthole-loaded GelMA shows better therapeutic potential in CIA mice model. (**A**) Experimental protocol for CAIA induction. (**B**) The hind ankle joints of CIA mice were subjected to H&E staining (20×), and the bone microstructures were evaluated using micro-CT. Relative levels of IL-6 (**C**), IL-1β (**D**), miR-1224-3p (**E**), and AGO1 (**F**) were measured by qPCR in mice synovial tissues. All statistical data are represented as mean ± SD (*n* = 6, * *p* < 0.05, ** *p* < 0.01).

**Figure 7 ijms-24-13210-f007:**
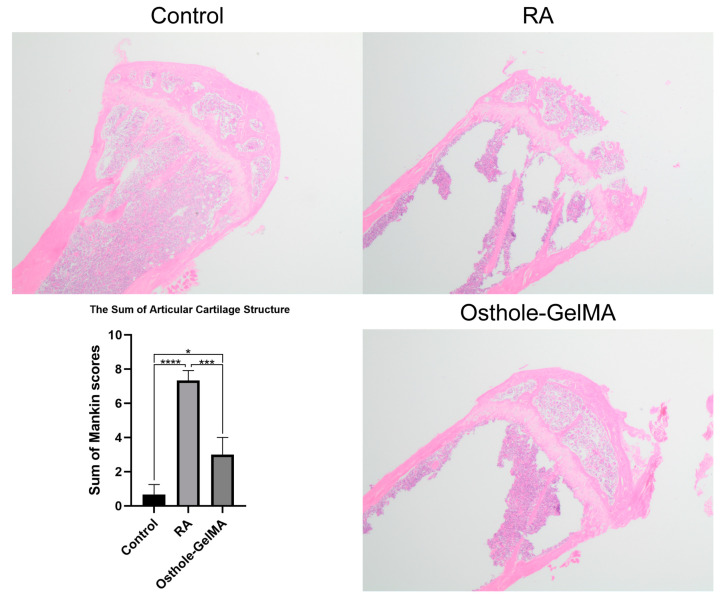
H&E staining was used to assess the effect of osthole-loaded GelMA on CIA mouse model. (4×) All statistical data are represented as mean ± SD (*n* = 6, * *p* < 0.05, *** *p* < 0.001, **** *p* < 0.0001).

**Figure 8 ijms-24-13210-f008:**
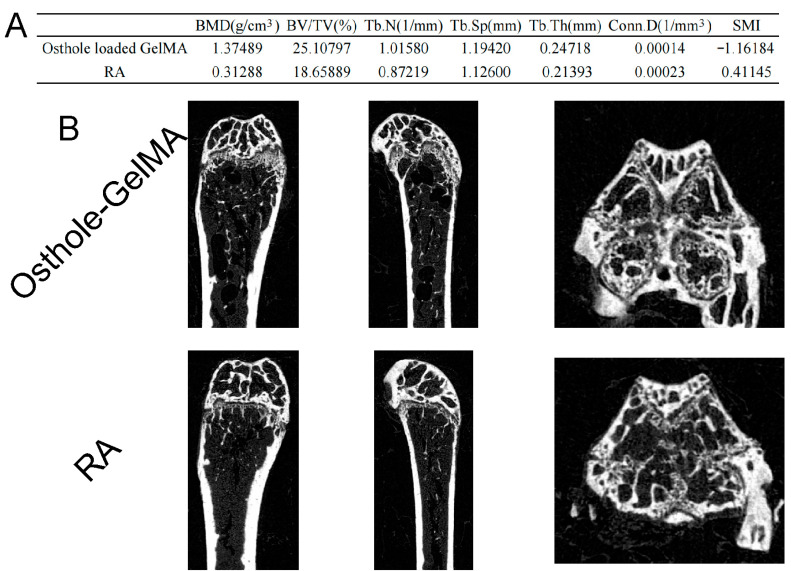
Micro-CT to assess the effect of osthole-loaded GelMA in CIA mice model. (**A**) Data such as femoral head scanning bone mineral density were evaluated for differences. (**B**) Sagittal, coronal, and axial views of the femoral heads of mice in different experimental groups.

## Data Availability

The study data have been contained within the manuscript and Appendix A.

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
