# Peer review of "GelMA Hydrogel as a Promising Delivery System for Osthole in the Treatment of Rheumatoid Arthritis: Targeting the miR-1224-3p/AGO1 Axis"

_ijms, 2023, doi:10.3390/ijms241713210_

Round 1
Reviewer 1 Report
lease clarify the core of the PAMAM dendrimer used in the present study (ethylendiamine or 1,4 diaminobutane or...). The core of the PAMAM dendrimer may affect the properties of the dendrimer for gene delivery as well as its toxicity. lease clarify the core of the PAMAM dendrimer used in the present study (ethylendiamine or 1,4 diaminobutane or...). The core of the PAMAM dendrimer may Please clarify the suitable concentration used in the present study and what is the reaseon used gelma compared to other natural polymers that have been used
Author Response
Dear editor:
Many thanks for your comments,but The content of this comment is not relevant to our work. In our work, we didn’t use dendrimer, we think this comment is a mistake. will you please confirm again.
best wishes
Jingsong Wei
Reviewer 2 Report
In this study, the authors evaluated the function of miR-1224-3p which show reduced expression in synovial tissues of rheumatoid arthritis (RA) patients compared to samples from osteoarthritis patients. The authors report that miR-1224-3p regulated AGO1 expression in primary RA synovial fibroblasts and that reduced miR-1224-3p expression correlated with increased expression of AGO1 expression and basal cytokine secretion (IL-6, IL-1b). Furthermore, the authors provide evidence that treatment of RA synovial fibroblasts with osthole, an antioxidant and anti-inflammatory compound extracted from Cnidium monnieri, increased miR-1224-3p levels. Osthole-mediated miR-1224-3p expression was associated with lower AGO1 protein levels, reduced expression of cytokines, and decreased synovial fibroblast proliferation. Osthole loaded in GelMA, a photo-crosslinked gelatin frequently used for biomedical applications, more efficiently decreased AGO1 expression in RA synovial fibroblast than osthole. Finally, osthole-loaded in GelMA shows better therapeutic effects in the mouse model of collagen-induced arthritis.
General comments: The authors provide a strong rationale for their experimental approaches. The data are convincing. I have a few issues with the writing and data interpretation. The authors must strengthen the materials and methods.
Specific points:
Abstract, line 12: We evaluated the levels…
Abstract, line 14: joint morphology
Abstract, line 16: The data showed…
Page 4, lines 146-147: …, HUM-iCell-s010 RA cells were treated with 50μM of osthole.
Page 4, lines 148-149: …the changes in miR-1224-3p and Ago1 levels.
Page 4, line 152: proliferation
Page 4, lines 151-154 and Fig. 4: The text show be revised. The wound healing assay is inappropriate for monitoring cell proliferation. The wound-closing assay cannot distinguish whether osthole inhibits synovial fibroblast proliferation or migration. A more specific assay is required to measure cell proliferation. The MTT assay kit is one of the most widely used assays for measuring cell proliferation.
Figs. 2-5. All figure legends must provide information on the number of experiments (n=?) and how data are represented (mean ± SE?).
Fig. 3, the legend: Please provide the time mRNA, protein, and cell are used post-osthole treatment.
Fig. 4: How was the fluorescence intensity ratio monitored? Please provide details in the figure legend (or materials and methods). A control with GelMA alone would strengthen the data.
Page 7, line 204: Treatment with osthole is on day 10 following the second CIA injection.
Page 7, line 217: However, compared to osthole treatment, the groups treated with
Page 8, line 236: HE
Paragraph 4.2: Please indicate whether the HUM-iCell-s010 and HUM-iCell-s010RA cell lines are knee synovial fibroblasts. The authors must provide the passage numbers for the cell lines.
Page 11, line 242: 500 µL
Paragraph 4.10: Writing the technique as a bullet point list is unacceptable.
Paragraph 4.11: Please revise this section. I understand immunofluorescence staining was for the cultured synovial fibroblasts and not synovial tissues.
English can be improved.
Author Response
Dear editor:
Many thanks for your comments . I had revised my paper according to your comments.
best wishes
Jingsong Wei
Specific points:
Abstract, line 12: We evaluated the levels…
Answer:Thanks for your careful check. We have revised the sentence in manuscript and marked as red.
Abstract, line 14: joint morphology
Answer:Thanks for your careful check. We have revised the sentence in manuscript and marked as red.
Abstract, line 16: The data showed…
Answer:Thanks for your careful check. We have revised the sentence in manuscript and marked as red.
Page 4, lines 146-147: …, HUM-iCell-s010 RA cells were treated with 50μM of osthole.
Answer:Thanks for your careful check. We have revised the sentence in manuscript and marked as red.
Page 4, lines 148-149: …the changes in miR-1224-3p and Ago1 levels.
Answer:Thanks for your careful check. We have revised the sentence in manuscript and marked as red.
Page 4, line 152: proliferation
Answer:Thanks for your careful check. We have revised the sentence in manuscript and marked as red.
Page 4, lines 151-154 and Fig. 4: The text show be revised. The wound healing assay is inappropriate for monitoring cell proliferation. The wound-closing assay cannot distinguish whether osthole inhibits synovial fibroblast proliferation or migration. A more specific assay is required to measure cell proliferation. The MTT assay kit is one of the most widely used assays for measuring cell proliferation.
Answer:Thanks for your rigorous comment, we have changed the proliferation of HUM-iCell-s010 RA cell from wound healing assay to MTT assay. The result was shown in the revised Fig. 3. The experimental procedure has added to the paragraph 4.9, and the text was marked as red.
Figs. 2-5. All figure legends must provide information on the number of experiments (n=?) and how data are represented (mean ± SE?).
We appreciate for your valuable comment. More description about the number of experiments and significant difference of the Figs. 2-5 has been added in the revised manuscript.
Fig. 3, the legend: Please provide the time mRNA, protein, and cell are used post-osthole treatment.
Answer:Thanks for your rigorous advice. The HUM-iCell-s010RA cells was treated with 50μM of osthole for 24h, and then he IL-6, IL-1β and miR-1224-3p was extracted according to instructions in manufacture’s kit (RNAiso Plus). According to your advice, the treat time was added and marked as red to the legend and line 148 in manuscript.
Fig. 4: How was the fluorescence intensity ratio monitored? Please provide details in the figure legend (or materials and methods). A control with GelMA alone would strengthen the data.
Answer:Thank you for your rigorous advice. The fluorescence intensity ratio was measured by ImageJ. GelMA has great biocompatibility and many works showed that GelMA hydrogel will promote cell proliferation (Choi B Y, Chalisserry E P, Kim M H, et al. The influence of astaxanthin on the proliferation of adipose-derived mesenchymal stem cells in gelatin-methacryloyl (GelMA) hydrogels[J]. Materials, 2019, 12(15): 2416.). So, we believe that GelMA alone will promote the growth state of HUM-iCell-s010RA cells. And in this study, GelMA act as a carrier platform, so the effect of GelMA with Osthole is the major research objective.
Page 7, line 204: Treatment with osthole is on day 10 following the second CIA injection.
Answer:Thanks for your careful check. We have revised the sentence in manuscript and marked as red.
Page 7, line 217: However, compared to osthole treatment, the groups treated with
Answer:Thanks for your careful check. We have revised the sentence in manuscript and marked as red.
Page 8, line 236: HE
Paragraph 4.2: Please indicate whether the HUM-iCell-s010 and HUM-iCell-s010RA cell lines are knee synovial fibroblasts. The authors must provide the passage numbers for the cell lines.
Answer:Thanks for your question. Firstly, HUM-iCell-s010 and HUM-iCell-s010RA cell lines are knee synovial fibroblasts. As for the passage numbers, the HUM-iCells-s010 and HUM-iCell-s010RA cell were expanded up to 3 passages for further experiment. And the information of cell passage was added to the manuscript: “Cells were expanded up to 3 passages for further use.” And was marked as red.
Page 11, line 242: 500 µL
Answer:Thanks for your careful check. We have revised the sentence in manuscript and marked as red.
Paragraph 4.10: Writing the technique as a bullet point list is unacceptable.
Answer:Thanks for your advice. We have changed the subtitle to “Histological evaluation”.
Paragraph 4.11: Please revise this section. I understand immunofluorescence staining was for the cultured synovial fibroblasts and not synovial tissues.
Answer:Thanks so much for your careful check. We have revised the text in manuscript.
Comments on the Quality of English Language English can be improved.
Answer:Thanks so much for your careful check. We have improved the text in manuscript.
Round 2
Reviewer 1 Report
Thank you. it should be fine.
Reviewer 2 Report
General Comments: I thank the authors for their reply and for having addressed most of the reviewer’s comments. The authors strengthened their manuscript. However, there are still minor issues that deserve attention.
Paragraph 4.10: As already started in the first round of review, it is unacceptable writing a technique in the form of bullet points. The authors must strengthen this paragraph.
Paragraph 4.13: Please indicate the age of the mice and the number of mice per group.
Fig. 6, lane 250: CIA induction.
Figs. 6-7. As for other figures, please provide information on the number of experiments/mice (n=?) and how data are represented (mean ± SD).
Fig. 7, Legend: What do the symbols *, ***, and **** mean?
Page 11, line 365: The resultant 500 mL of transfection solutions were…
Overall, the English language is fine. There are a few typos.
Author Response
Paragraph 4.10: As already started in the first round of review, it is unacceptable writing a technique in the form of bullet points. The authors must strengthen this paragraph.
Answer: We have changed the description of this paragraph in revised manuscript and marked as red.
Paragraph 4.13: Please indicate the age of the mice and the number of mice per group.
Answer: I had added the information in the main text by red text.
Fig. 6, lane 250: CIA induction.
Answer: I had revised the text.
Figs. 6-7. As for other figures, please provide information on the number of experiments/mice (n=?) and how data are represented (mean ± SD).
Answer:I had added the information and revised the wrong text.
Fig. 7, Legend: What do the symbols *, ***, and **** mean?
Answer:I had added the information.
Page 11, line 365: The resultant 500 mL of transfection solutions were…
Answer:I had revised the mistaken text.